# Impact of Work–Life Balance on the Quality of Life of Spanish Nurses during the Sixth Wave of the COVID-19 Pandemic: A Cross-Sectional Study

**DOI:** 10.3390/healthcare12050598

**Published:** 2024-03-06

**Authors:** Ana María Antolí-Jover, María Adelaida Álvarez-Serrano, María Gázquez-López, Adelina Martín-Salvador, María Ángeles Pérez-Morente, Encarnación Martínez-García, Inmaculada García-García

**Affiliations:** 1Department of Nursing, Faculty of Health Sciences, University of Granada, 51001 Ceuta, Spain; antolijover@ugr.es; 2Department of Nursing, Faculty of Health Sciences, University of Granada, 18016 Granada, Spain; ademartin@ugr.es (A.M.-S.); emartinez@ugr.es (E.M.-G.); igarcia@ugr.es (I.G.-G.); 3Department of Nursing, Faculty of Health Sciences, University of Jaén, 23071 Jaén, Spain; mmorente@ujaen.es; 4Virgen de las Nieves University Hospital, 18014 Granada, Spain

**Keywords:** health-related quality of life, EQ-5D, work–life balance, nurse, COVID-19

## Abstract

This study addresses the health-related quality of life (HRQoL) of Spanish nurses during the sixth wave of the COVID-19 pandemic, assessed through the EQ-5D and EQ-VAS indices. Methods: This cross-sectional 334 study used online surveys, recruiting 305 Spanish nurses. Results: Nurses generally perceived a good HRQoL. “Negative work–family interaction” is adversely associated with the EQ-VAS (β = −0.337, 95% CI [−1.733, −0.723]) and EQ-5D (β = −0.399, 95% CI [−0.021, −0.01]) indices, while “positive work–family interaction” shows a positive relationship with the EQ-VAS (β = 0.218, 95% CI [0.381, 1.759]). The presence of a “paid supportive caregiver” is positively associated with the EQ-VAS (β = 0.18, 95% CI [1.47, 12.3]) and EQ-5D (β = 0.149, 95% CI [0.004, 0.117]) indices, but a higher “number of children” is negatively linked with the EQ-5D index (β = −0.146, 95% CI [−0.061, −0.002]). In addition, living with a partner (EQ-VAS β = 0.16, 95% CI [1.094, 14.67] and EQ-5D index β = 0.174, 95% CI [0.018, 0.163]) and working a “rotating shift” (EQ-5D index β = 0.158, 95% CI [0.005, 0.098]) are positively associated. Conclusions: These findings highlight the need to comprehensively address nurses’ well-being, considering both their working conditions and their home environment, especially in crisis contexts such as the current pandemic.

## 1. Introduction

Health status and health-related quality of life (HRQoL) have been used synonymously since health was defined by the World Health Organisation (WHO) in 1948 [1]. Despite discussions in 2011 on the need to update this definition to reflect the increase in chronic diseases, the official definition has not been changed [2]. Calman, in 1984, proposed one of the first definitions of HRQoL because of the need to distinguish between the two, and to focus health care on the person [3]. According to Calman, health-related quality of life is “the difference over a specified period of time between the individual’s hopes and expectations and actual experiences”. Therefore, HRQoL refers to how a person’s experiences in terms of health and well-being compare with their expectations and hopes.

The general perception among health professionals and the active attitude of patients have devalued the results obtained by traditional health metrics and propelled the clinical investigation of HRQoL [1,4], which has been consolidated as a term that refers to the perception and evaluation that a person has about their overall well-being without being solely about the absence of disease. The feelings that a person experiences have been integrated into three dimensions: physical, emotional and social [5]. Health-related quality of life (HRQoL), therefore, focusses on assessing the subjective influence of health status, health care, prevention and health promotion activities on an individual’s ability to achieve and maintain a level of functioning that enables the achievement of life goals, and is reflected in overall well-being [6]. Inquiring into the HRQoL of health professionals is very important, given that research postulates that declines in the mental and physical health of health professionals are associated with errors in clinical practice and symptoms of depression in health professionals [7,8]. Different studies indicate that the deterioration of health professionals’ health, in addition to limiting their ability to provide high quality care, influences the weakening of care services and increases patient morbidity and mortality [9,10].

This connection between the HRQoL of health care professionals and the HRQoL of patients becomes a key component in understanding and improving health outcomes in the general population [11,12]. As a result of this, the exploration of those factors that affect the HRQoL of health care professionals is a fundamental aspect. A review of the literature indicates that age, gender, shift work, work environment, job satisfaction and work–life balance significantly impact nurses’ HRQoL [12,13,14,15,16].

Work–life balance has gained great importance in organisational health studies, especially among nurses. It has been observed that conflict or imbalance in both domains leads to increased stress and burnout [17]. Furthermore, there is evidence that nurses who spend more time at work than in their private lives experience job dissatisfaction and poor quality of life [14,18]. This lack of balance, therefore, not only affects the well-being of nurses, but may also impact patient safety and health outcomes [11,14].

The COVID-19 pandemic triggered a challenging and unprecedented episode that disrupted the life balance of nursing professionals. This health crisis generated a high demand for health services. The lack of previous scientific knowledge in this area caused significant difficulties in most countries worldwide [19,20]. This resulted in an increased demand for nursing care from the population, with the implied difficulty of providing adequate care [21]. Thus, health care workers were exposed to highly stressful situations throughout the pandemic [20], combining occupational and non-occupational risk factors such as insecurity, confinement measures, extensive media coverage and fear of contamination [22]. In addition, the pandemic also cast a shadow of discrimination over health workers, including nurses, adding another layer of complexity and emotional burden to their already challenging circumstances [23]. The stigma associated with being on the front lines of fighting a highly contagious disease not only exacerbated the stress and emotional toll, but also created barriers to social acceptance and support for these professionals. Nurses found themselves confronting societal prejudices and misconceptions about the risks and realities of their profession, which further increased their psychological and emotional challenges [24].

All of these had a negative impact, as evidenced by higher rates of anxiety, depression and stress among nursing professionals [25,26,27]. In addition, this presented challenges to the workers in adjusting their own emotional regulation mechanisms and adaptive capacities to maintain optimal mental health [20]. Although there is extensive literature addressing mental health and specific physical illnesses, surprisingly few studies have explored the direct effect of the pandemic on nurses’ work–life balance or HRQoL.

Therefore, there is a need for health care organisations to recognise the critical importance of work–life balance, and to assess the impact on the HRQoL of health care professionals.

The purpose of this research is to describe the quality of life of nursing professionals in Spain during the sixth wave of the COVID-19 pandemic, and to also evaluate the influence of sociodemographic and occupational variables as well as interactions between work and family on the health-related quality of life of these professionals.

## 2. Materials and Methods

### 2.1. Study Design and Population

This cross-sectional study was conducted during November and December 2021 using an online survey. Participants were recruited from nurses working in hospitals and other health care facilities, both public and private. None of the workers included in this research were working remotely.

The study subjects were recruited from both mainland Spain and non-mainland Spain (Canary Islands, Balearic Islands and the autonomous cities of Ceuta and Melilla).

The sampling method used was “snowball sampling”, carried out through an online platform that recruited participants by disseminating an invitation link. In order to extend the coverage and ensure the validity of this sampling approach, the active collaboration of the Spanish Nursing Associations and the SATSE Union was sought. These entities played an important role in disseminating the survey among their members and affiliates through their websites and social networks. To facilitate this process, a letter introducing the project, including a link to the survey, was drafted and emailed to potential participants. This “snowball” approach allowed a diverse and representative sample of nurses to be reached, using existing networks at the collegiate and union levels to maximise participation.

To calculate the sample size, the number of registered nurses in 2020 according to the National Institute of Statistics was consulted and estimated at 330,745. A minimum sample size was determined for a confidence level of 95% and a margin of error of 6%, resulting in a total of 267 nurses.

### 2.2. Procedures and Measures

A self-administered questionnaire was designed, consisting of two sections that could be completed in a maximum of twenty minutes.

In the first section, the socio-demographic characteristics of the sample and occupational factors were investigated: age of the participants (in years), sex, geographical location (peninsular or non-peninsular), having been diagnosed with COVID-19, having been in isolation due to close contact with a confirmed case of COVID-19, religion (with religious beliefs, with no religious beliefs), relationship status (living with partner or living without a partner), number of children and whether they had the help of a paid caregiver to care for them. In addition, length of service (less than 5 years or more than 5 years), whether they worked rotating shifts, whether they were assigned to night shifts and whether they held a position of managerial responsibility were assessed.

The second section included two standardised questionnaires: the European Quality of Life Questionnaire-5 Dimensions (EQ-5D) and the Work–Family Interaction Questionnaire (SWING).

The European Quality of Life Questionnaire-5 Dimensions (EQ-5D) is a widely used instrument developed in Europe to assess key aspects of quality of life. This tool assesses mobility, self-care ability, daily activities, pain/discomfort and anxiety/depression through one question for each of these five dimensions. Using an algorithm, the answers provided allow the calculation of the EQ-5D index, where 0 represents a state equivalent to death and 1 indicates perfect health. In addition, the EQ-5D questionnaire includes a visual analogue scale (VAS) that assesses respondents’ perception of health on a scale from 0 (representing the worst imaginable well-being) to 100 (indicating the best imaginable well-being). Specifically, the EQ-5D index value reflects health status, while the EQ-VAS provides information on individual perception of health [28]. This questionnaire was validated for the Spanish population [29], since the EQ-5D has been tested in numerous studies to verify its psychometric properties (validity, reliability, sensitivity to change) [30,31,32].

The Work–Family Interaction Questionnaire (SWING): The negative work–family interaction test consists of 8 items, with a Likert-type response format with 4 response options (from 0 to 3). In it, the person indicates the frequency with which he/she perceives each of the situations proposed in the questionnaire.

The negative family–work interaction test consists of 4 items, and has the same format as the negative work–family interaction test.

The positive work–family interaction test and the positive work–family interaction test each consist of 5 items. They use the same response format as the previous tests.

This questionnaire, validated for the Spanish population by Moreno and Sanz (2009), has shown robust internal consistency, evidenced by Cronbach’s alpha values between 0.77 and 0.89 [33]. To facilitate the interpretation of the results, scores between 0 (inclusive) and less than 1 point are defined as “low interaction”; scores between 1 point (inclusive) and less than 2 points are defined as “medium interaction”; and scores between 2 points (inclusive) and up to 3 points are defined as “high interaction”.

### 2.3. Ethical Issues

This study complies with the good clinical practice regulations, as stated in the European Directive 2001/20/CE and Law 14/2007 of 3 July on biomedical research. Treatment of personal data in health research is governed by Organic Law 3/2018 of 5 December on Data Protection and Guarantee of Digital Rights. The protocol obtained a favourable resolution from the Biomedical Research Ethics Committee of the province of Granada, and was approved by the Dean of the Faculty of Health Sciences of Ceuta on the 13th of September of 2021. Participants were informed of the objectives of the study and provided their consent to participate by checking a specific box.

### 2.4. Statistical Analysis

In the statistical analysis of the data, descriptive analyses were carried out for all variables, expressing categorical variables in terms of frequency and proportion, and continuous variables as means and standard deviations. After applying the Kolmogorov–Smirnov test, the parametric nature of the data was confirmed, allowing for a parametric bivariate analysis.

To assess differences between groups, Student’s *t*-tests for independent samples were used. In addition, the effect size was calculated using Cohen’s d coefficient. Pearson’s correlation was used to assess relationships between continuous variables. Finally, in order to identify possible explanatory factors for the results analysed in this research, two multiple linear regression models were carried out: one for the EQ-5D index and the other for the EQ-VAS.

Values of *p* < 0.05 in all tests were considered significant. Statistical analysis was performed using IBM SPSS Statistics for Macintosh, Version 25.0. (IBM Corp, Armonk, NY, USA).

## 3. Results

### 3.1. Sample Characteristics

Table 1 presents the sociodemographic characteristics of the sample composed of 305 study participants, 38 more than the minimum estimated. The mean age of the participants was 38.8 years, with a standard deviation of 11.383 years.

In relation to gender, it was observed that 13.1% of the participants were male, while 86.8% were female. In terms of geographical distribution, the majority of participants (76.1%) were from the mainland, while the remaining 23.9% were from outside the mainland, including Ceuta, Melilla and the Balearic and Canary Islands.

Regarding the diagnosis of COVID-19, 17.7% had been diagnosed with the disease. Finally, regarding isolation due to close contact with confirmed COVID-19 cases, 33.8% of participants had experienced this situation.

Regarding religious beliefs, 42.6% of the participants claimed to have religious beliefs.

Regarding marital status, 74.7% of the participants lived with a partner, while 25.2% lived without a partner. Regarding parenthood, 57.4% of the participants had children, of which 11.5% had a paid caregiver for their care, while the remaining 88.5% did not make use of this service.

In addition, data were collected on the number of children per participant, yielding an average of 1.03 children, with a standard deviation of 1.062. The average age of the youngest child was 11.29 years, with a standard deviation of 9.516 years, while the average age of the oldest child was 13.78 years, with a standard deviation of 10.492 years.

In the employment context of the participants (Table 2), the following characteristics were observed. Regarding length of service, 55.1% had less than 5 years of experience, while 44.9% had more than 5 years of service. In addition, 59.7% worked on a rotating shift system, and 52.8% worked night shifts. In terms of management roles, 16.1% of the participants held a position of responsibility.

Regarding work–life balance, the following values were recorded from lowest to highest: negative work–family interaction with a mean score of 0.46 (SD 0.443); negative work–family interaction with a mean score of 1.24 (SD 0.519); positive work–family interaction with a mean score of 1.44 (SD 0.634) and positive work–family interaction with a mean score of 1.95 (SD 0.693).

As for the quality of life indicators in the overall sample, an EQ-5D index of 0.820 (S.D. 0.154) and an EQ-VAS of 74.56 (S.D. 15.735) were recorded.

### 3.2. Bivariate Analysis

Table 3 presents the results of the bivariate analysis of the EQ-5D index and the EQ-VAS in relation to several sociodemographic variables.

Significant differences are observed by sex in relation to the EQ-VAS, revealing that women have a significantly lower score compared to men (*p* = 0.008, d = 0.46). Furthermore, a positive COVID-19 diagnosis is associated with a significantly lower EQ-5D index compared to those who did not receive such a diagnosis (*p* = 0.042, d = 0.306).

Table 4 reflects the bivariate analysis of the work variables and EQ-5D index and EQ-VAS, with the variables related to work–life balance being those that reached statistical significance.

The negative work–family and family–work interactions maintain an inversely proportional relationship with the EQ-5D index (*p* < 0.001) and the EQ-VAS (*p* < 0.001).

In contrast, the positive work–family interaction is positively related to the EQ-5D index (*p* = 0.02) and EQ-VAS (*p* = 0.009).

### 3.3. Multivariate Analysis

Table 5 presents the final multiple linear regression model for the EQ-5D index. When all the variables were introduced, the model was composed of five variables that explain 20% of the variability in the EQ-5D index scores.

In particular, the variable “Negative work–family interaction” showed a significant negative association (β = −0.399, 95% CI [−0.021, −0.01]), indicating that negative experiences related to work–family dynamics were related with lower scores on the EQ-5D index. On the other hand, having a paid caregiver to care for children showed a significant positive association (β = 0.149, 95% CI [0.004, 0.117]), suggesting that delegating childcare to a paid caregiver has a positive impact on the EQ-5D index. Furthermore, the number of children exhibited a significant negative association (β = −0.146, 95% CI [−0.061, −0.002]), suggesting that a greater number of children was related to lower scores on the EQ-5D index. Likewise, “living with a partner” (β = 0.174, 95% CI [0.018, 0.163]) and “working a rotating shift” (β = 0.158, 95% CI [0.005, 0.098]) also presented significant positive associations, indicating that living as a couple and working in rotating shifts were related to higher scores on the EQ-5D index.

Table 6 presents the final linear regression model for the EQ-VAS. By introducing all variables, the model identified four predictor variables that together explain 21% of the variability in EQ-VAS scores.

Specifically, the variable “Negative work–family interaction” demonstrated a significant negative association (β = −0.337, 95% CI [−1.733, −0.723]) with the EQ-VAS, suggesting that negative experiences related to work–family dynamics were associated with lower EQ-VAS scores. Likewise, “Positive work–family interaction” showed a positive association (β = 0.218, 95% CI [0.381, 1.759]), indicating that positive aspects of work–family interactions were related to higher EQ-VAS scores. Furthermore, having a paid caregiver to care for children (β = 0.18, 95% CI [1.47, 12.334]) and living with a partner (β = 0.16, 95% CI [1.094, 14.67]) were associated positively with EQ-VAS scores.

## 4. Discussion

This study focussed on assessing the quality of life of Spanish nursing professionals during the so-called sixth wave of the COVID-19 pandemic in Spain. In addition, the relationship between quality of life and sociodemographic and occupational variables, as well as work–family interactions, were studied.

The EQ-5D and EQ-VAS values obtained showed good health status and perceived quality of life. These results could be explained by the low incidence of COVID-19 cases in this wave (with only 17.7% of positive cases and 33.8% of strict isolation) compared to the figures at the beginning of the pandemic [34]. It is also plausible that participants had already adapted to the magnitude of the pandemic, and that improvements in the effective management of the situation had already materialised [19]. These changes may have influenced their perceptions, resulting in more resilient responses compared to the start of the wave, as noted by several authors [35,36,37,38].

In this context, the study by Baysal et al. is relevant, as it established a connection between fear of COVID-19 and subscales of nurses’ professional quality of life [37]. This finding provides a basis for understanding how psychological and emotional aspects related to the pandemic may influence quality of life.

On the other hand, the EQ-5D and EQ-VAS scores show differences according to the work and sociodemographic variables analysed, as well as with work–family interactions. With regard to the sociodemographic variables, women showed significantly lower scores on the EQ-VAS, suggesting that they perceived their quality of life as less satisfactory. This could be related to factors linked to cultural gender differences rooted in the double burden of caring for the family and running the household [39,40,41]. In this sense, Amezcua affirms that caring can lead to more difficulties in carrying out a full life with social, professional and family trajectories [21]. In fact, Salamanca (2019) highlights that it is much more difficult for female nurses to give up caring for others in order to become the ones cared for [42].

Regarding the national territory, although extra-peninsular participants presented higher scores in both indices (EQ-5D index and EQ-VAS), no statistically significant differences were found. However, according to data from the National Statistics Institute (INE), except for the Balearic Islands, the extra-peninsular regions have a quality of life index that is below the national average (IMCV_ccaaES, 2023). Given these discrepant results and the scarcity of specific studies in the field of nursing, it may be necessary to carry out further research to investigate this issue.

Three variables that are closely related will be discussed below. It could be said that the “number of children”, the presence of a paid caregiver for them and living with a partner are determining factors in the family environment. These elements significantly influence the dynamics and structure of a family, affecting both day-to-day responsibilities and interpersonal relationships within the household. Together, these elements contribute to defining the nature and functioning of the family environment [43] and, according to the results of our multivariate analysis, explain the variability of the EQ-5D index and EQ-VAS.

Our results reveal a negative relationship between the “number of children” and the EQ-5D index, supported by the multivariate analysis, where it is observed that as the number of children increases, the quality of life (EQ-5D index) decreases. Our results are consistent with studies such as that of Amezcua et al. (2020), who state that being part of a close, loving and supportive couple relationship creates a space for a better health-related quality of life [21]. According to this, having more children would represent a major obstacle to the well-being of the participants’ own health and quality of life [43]. However, other studies have shown that being a parent increases the feeling of happiness [44] with a meaning of importance in life as a fundamental human need in the evolutionary perspective.

In this sense, it would be desirable for company policies to take into account the numbers of children employees have. Thus, if necessary, adjustments could be made to workplaces to facilitate and improve the quality of life of workers, should the workers themselves request it.

The number of children can impact family dynamics, as each additional member may introduce logistical and emotional complexities. Consequently, the assistance of a paid caregiver could influence time management and family responsibilities [45]. Our findings support this possibility, as the presence of a “paid supportive caregiver” was positively associated with scores on both HRQoL indicators (EQ-5D index and EQ-VAS) according to our multivariate analysis.

Finally, in this family domain, living with a partner or living without a partner is crucial, as it establishes the basis of the family unit and affects the emotional and economic stability of the household [43]. Living “with a partner” also showed a significant positive association and explains an increase in both HRQoL indices (EQ-5D index and EQ-VAS). This indicates that living with a partner is related to a better perceived quality of life. Holt-Lunstad states that social relationships have powerful influences on health and longevity [46]. Zepeda and Sánchez consider that being part of a close, affectionate and supportive relationship with a partner creates a space for better quality of life and health by influencing attitudes, meanings and knowledge that determine the adoption of healthy behaviours [47].

While no significant differences were observed on the EQ-VAS, participants diagnosed with COVID-19 had significantly lower scores on the EQ-5D index compared to those without a diagnosis. The diagnosis of COVID-19 has been highlighted as an important factor, given the global influence of the pandemic on physical and mental health [48,49,50]. Cortéz’s research supports this connection by identifying greater somatisation among professionals who experienced personal infection or loss of a family member due to COVID-19 [51]. Similarly, Baiào et al. stated that the physical and psychological consequences of health care delivery during the pandemic negatively impacted the quality of life of nursing professionals, which is consistent with the results of our study [52].

On work factors, working a “rotating shift” indicates that participants working rotating shifts have higher quality of life scores (EQ-5D index). The reconciliation of work and family life are perceived as stressors, due to the incompatibility of work and school schedules. In this context, the implementation of on-demand shifts has been shown to significantly reduce absenteeism [53,54].

During research on work–family interactions, more emphasis was placed on exploring the conflict between the two roles. However, the importance of investigating the positive influences arising from the combination of work and family responsibilities has been recognised [54,55,56]. Therefore, in our study, both negative and positive interactions were considered. The results obtained show that positive influences are more relevant to the perception of work–life balance than negative influences, corroborating previous research such as that carried out by Velásquez and Tovar [55]. These data confirm that the adoption of a comprehensive approach that addresses both dimensions of work–family reconciliation allows for a more complete vision.

Finally, it is evident that the influence of work on family acts as a significant predictor of health-related quality of life. The relevance of family and employment as fundamental institutions in people’s lives is undeniable [56]. The negative association identified in our study between “negative work–family interaction” and health-related quality of life reveals how work demands can adversely impact work–family harmony, affecting people’s overall perception of well-being. Work–family conflict, according to Matarsat, Rahman and Abdul-Mumin (2021), negatively impacts workers’ quality of life (CVT) and their physical health [57]. Workers experience symptoms of psychological distress, including anxiety, depression, stress and burnout, as well as decreased life satisfaction [58,59]. This effect is attributed to the pressures derived from work and family roles, which affect the worker’s ability to manage his or her life satisfactorily. On the other hand, we should not underestimate the influence of the family role on work. Our results reveal an inversely proportional relationship with HRQoL. Authors such as Mauno and Roukolainen (2017), and subsequently Headrick et al. (2023), present results in this direction. They add that when recovery opportunities after work are inadequate, both in quantity and quality, due to high family demands, the biological system is altered and individuals face conflicts in the assumption of work and family roles, which may have adverse health consequences [60,61].

In contrast to the negative relationship between “positive work–family interaction” and health-related quality of life, our study reveals a positive association predicting an increase in quality of life (EQ-VAS) by increasing “negative work–family interaction”. This finding suggests that positive experiences in reconciling work and family are directly linked to a higher perception of quality of life. The empirical support provided by the study by Chan et al. (2020) reinforces this association, showing that both work–family enrichment (WFE) and family–work enrichment (FWE) are positively linked to job satisfaction, affective commitment, family satisfaction, as well as with physical and mental health [62].

The analysis models developed by authors such as Greenhaus, Carlson and Wang address the dynamics between work and family, focusing on effort–reward aspects [63,64,65]. These models suggest that effective conflict resolution in the workplace can have a positive impact on stress management in the family environment. The ability to handle stressful situations at work translates into benefits that enrich the family sphere. In this context, the ability to resolve work conflicts not only contributes to well-being at work, but also becomes a resource that strengthens family relationships by reducing tensions and worries that could negatively affect life at home. Effective conflict resolution at work, according to these models, acts as a facilitating factor that promotes a healthier balance between professional and family demands.

This dynamic not only affects the quality of life of nurses, but also has a direct impact on the quality of care offered to patients [66,67]. Addressing this balance is important to guide policies and make significant transformations in the field of health.

### Limitations

The present study has several limitations that require careful consideration when interpreting the results. First, due to the geographical dispersion of the participants, data collection was carried out using self-administered questionnaires. This form of collection allowed access to a more varied sample and, therefore, is more representative of social reality, especially considering the decentralisation of health competencies in Spain. However, it is crucial to recognise that this methodology may be affected by social desirability, which could influence responses and compromise measurement accuracy.

Furthermore, the adoption of a cross-sectional design limited the ability to follow-up over time, preventing causal relationships from being established and understanding the dynamics of change in nurses’ quality of life during the pandemic. Likewise, a lack of longitudinal studies carried out in different waves of the pandemic has been detected. This fact makes it difficult to evaluate the evolution in the quality of life of Spanish nurses, and to identify possible patterns of change in response to variations in working and social conditions. These limitations highlight the need for future research that addresses these aspects and provides a more complete and dynamic understanding of the quality of life of nurses in pandemic contexts, considering the logistical challenges and complexity of the realities of health care in Spain.

Despite these limitations, this study also presents several strengths that deserve to be highlighted. The use of self-administered questionnaires allowed access to a varied and representative sample of the social reality, providing a broad and diverse view of nurses’ quality of life during the pandemic. Furthermore, although the cross-sectional design limits the ability to follow up over time, it provides a valuable snapshot of the quality of life of these health professionals at a specific point in time during the pandemic, which is especially relevant given the constantly evolving nature of the COVID-19 pandemic.

Finally, this study highlights the need for future research that addresses these issues and provides a more comprehensive and dynamic understanding of nurses’ quality of life in pandemic contexts. This identified need may guide the direction of future research and contribute to improvements in working conditions and quality of life for nurses.

## 5. Conclusions

In the challenging context of the sixth wave of the pandemic, the purpose of this research was to describe the quality of life of nursing professionals in Spain during this COVID-19 timeframe. Additionally, we evaluated the influence of sociodemographic and occupational variables, as well as interactions between work and family, on the HRQoL of these professionals.

Our investigation highlights the intricate relationship between the personal and professional lives of nurses, emphasising the importance of considering family factors when assessing HRQoL. Through the analysis of indicators such as the presence of a paid caregiver, the number of children, the type of work shift and living with a partner, revealing patterns emerged.

We found that a higher “number of children” may present additional challenges in daily management, while the presence of a “paid supportive caregiver” is positively associated with a greater perceived quality of life. Furthermore, it is interesting to note that “cohabiting with a partner” and working in a “rotating shift” are associated with a more positive perception of HRQoL.

These findings underscore the need for workplace policies that recognise and address the complexities of nurses’ family lives. When designing work–life balance strategies, it is essential to consider how family and work aspects intertwine in daily development. It is time to advocate for policies that consider and support the diverse roles that nurses play both within and outside the workplace.

Ultimately, our study not only provides detailed insights into the HRQoL of nurses during the pandemic, but also prompts us to reflect on how we can create more compassionate and supportive work environments for those who dedicate their lives to caring for others.

## Figures and Tables

**Table 1 healthcare-12-00598-t001:** Characteristics of the sample: sociodemographic variables and COVID-19.

	Participants (305)
	M (SD)
**Age (years old)**	38.8 (11.383)
	**n (%)**
**Sex**	
Man	40 (13.1)
Woman	265 (86.8)
**Population**	
Peninsular	232 (76.1)
Extrapeninsular	73 (23.9)
**COVID-19 Diagnosis**	
Yes	54 (17.7)
No	251 (82.3)
**Close contact isolation**	
Yes	103 (33.8)
No	202 (66.2)
**Religion**	
With religious beliefs	130 (42.6)
With no religious beliefs	175 (57.4)
**Relationship status**	
Live with a partner	228 (74.7)
Live without a partner	77 (25.2)
**Children**	
Yes	175 (57.4)
No	130 (42.6)
**Paid Caregiver**	
Yes	35 (11.5)
No	270 (88.5)
	**M (SD)**
**Number of children**	1.03 (1.062)
Youngest son age	11.29 (9.516)
Oldest son age	13.78 (10.492)

M = mean; SD = standard deviation; n = frequency; % = percentage.

**Table 2 healthcare-12-00598-t002:** Characteristics of the sample: work-related variables, work–life balance and HRQoL.

	Participants
	n (%)
**Length of service**	
<5 years	168 (55.1)
>5 years	137 (44.9)
**Rotating shift**	
Yes	182 (59.7)
No	123 (40.3)
**Nocturnality**	
Yes	161 (52.8)
No	144 (47.2)
**Management**	
Yes	49 (16.1)
No	252 (82.6)
**Work–Life Balance**	**M (SD)**
Negative work–family interaction	1.24 (0.519)
Negative family–work interaction	0.46 (0.443)
Positive work–family interaction	1.44 (0.634)
Positive family–work interaction	1.95 (0.693)
**Quality of life**	
EQ-5D index	0.820 (0.154)
EQ-VAS	74.56 (15.735)

M = mean; SD = standard deviation; n = frequency; % = percentage.

**Table 3 healthcare-12-00598-t003:** Bivariate analysis of EQ-5D index and EQ-VAS by sociodemographic variables.

	EQ-5D Index			EQ-VAS		
	M (SD)	*p*	d	M (SD)	*p*	d
**Participants**	0.820 (0.154)			0.560 (0.735)		
**Sex**						
Woman	0.813 (0.152)	0.052		73.630 (15.976)	0.008 *	0.46
Man	0.864 (0.164)			80.730 (12.547)		
**Population**						
Peninsular	0.816 (0.150)	0.393		73.66 (15.34)	0.073	
Extrapeninsular	0.834 (0.169)			77.44 (16.715)		
**COVID-19 Diagnosis**						
Yes	0.782 (0.159)	0.042 *	0.306	72.48 (15.928)	0.284	
No	0.828 (0.152)			75.01 (15.689)		
**Close contact isulation**						
Yes	0.827 (0.149)	0.566		73.56 (15.401)	0.429	
No	0.817 (0.157)			75.07 (15.916)		
**Religion**						
With religious beliefs	0.827 (0.155)	0.282		75.44 (14.681)	0.189	
With no religious beliefs	0.807 (0.153)			72.96 (17.454)		
**Relationship status**						
Live with a partner	0.823 (0.156)	0.592		75.01 (15.217)	0.392	
Live without a partner	0.812 (0.150)			73.23 (17.215)		
**Son**						
Yes	0.822 (0.168)	0.825		75.190 (15.634)	0.418	
No	0.818 (0.134)			73.720 (15.891)		
**Paid Caregiver**						
Yes	0.862 (0.148)	0.086		78.66 (14.49)	0.102	
No	0.815 (0.154)			74.03 (15.84)		
	r	*p*		r	*p*	
**Age (years old)**	−0.039	0.495		−0.044	0.444	
**Number of children**	−0.057	0.324		0.015	0.8	
**Youngest son age**	−0.022	0.773		0.027	0.722	
**Oldest son age**	−0.032	0.672		0.003	0.969	

M = mean; SD = standard deviation; r = Pearson; * *p* < 0.05; d = Cohen’s d.

**Table 4 healthcare-12-00598-t004:** Bivariate analysis of EQ-5D index and EQ-VAS by work-related variables and work–life balance.

	EQ-5D Index			EQ-VAS		
	M (SD)	*p*	d	M (SD)	*p*	d
**Length of service**						
<5 years	0.828 (0.154)	0.354		75.36 (14.618)	0.327	
>5 years	0.811 (0.155)			73.58 (17.009)		
**Rotating shift**						
Yes	0.821 (0.151)	0.873		73.40 (15.69)	0.117	
No	0.818 (0.159)			76.28 (15.71)		
**Nocturnality**						
Yes	0.822 (0.152)	0.826		73.8 (15.55)	0.368	
No	0.818 (0.157)			75.42 (15.95)		
**Management**						
Yes	0.800 (0.169)	0.323		78.02 (16.076)	0.088	
No	0.824 (0.152)			73.81 (15.655)		
**Work–Life Balance**	r	*p*		r	*p*	
Negative work–family interaction	−0.388	0.0001 **		−0.376	0.0001 **	
Negative family–work interaction	−0.27	0.0001 **		−0.251	0.0001 **	
Positive work–family interaction	0.133	0.02 *		0.149	0.009 **	
Positive family–work interaction	0.014	0.809		0.019	0.736	

M = mean; SD = standard deviation; * *p* < 0.05; ** *p* < 0.00; d = Cohen’s d.

**Table 5 healthcare-12-00598-t005:** Multiple linear regression model for EQ-5D index.

Variable	β	Dev Error	95% CI	
Negative work–family interaction	−0.399	0.003	−0.021	−0.01
Paid support caregiver	0.149	0.029	0.004	0.117
Number of children	−0.146	0.015	−0.061	−0.002
Living with a partner	0.174	0.037	0.018	0.163
Rotating shift	0.158	0.024	0.005	0.098

Durbin–Watson test = 1.914; F = 8.464; *p* < 0.001.

**Table 6 healthcare-12-00598-t006:** Multiple linear regression model for EQ-VAS.

Variable	β	Error Dev.	95% CI	
Negative work–family interaction	−0.337	0.256	−1.733	−0.723
Positive work–family interaction	0.218	0.349	0.381	1.759
Paid support caregiver	0.18	2.751	1.47	12.334
Living with a partner	0.16	3.438	1.094	14.67

Durbin–Watson test = 1.767; F = 11.008; *p* < 0.023.

## Data Availability

The data presented in this study are available on request from the corresponding author. The data are not publicly available due to privacy.

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
