# Peer review of "Impact of Work–Life Balance on the Quality of Life of Spanish Nurses during the Sixth Wave of the COVID-19 Pandemic: A Cross-Sectional Study"

_healthcare, 2024, doi:10.3390/healthcare12050598_

Round 1
Reviewer 1 Report
Comments and Suggestions for Authors
In general, the research should drive the design. But sometimes, the progression of the research helps determine which design is most appropriate. Your adoption of a Cross-sectional design is not a problem. But now that you have established whether there are links or associations between certain variables you must set up a longitudinal study to provide a more complete and dynamic understanding of the quality of life of nurses in pandemic contexts. Congratulations on your work.
Author Response
In general, the research should drive the design. But sometimes, the progression of the research helps determine which design is most appropriate. Your adoption of a Cross-sectional design is not a problem. But now that you have established whether there are links or associations between certain variables you must set up a longitudinal study to provide a more complete and dynamic understanding of the quality of life of nurses in pandemic contexts. Congratulations on your work.
Response: We sincerely appreciate your suggestion and value your interest in our work. We will take into consideration your recommendation regarding the implementation of a longitudinal study to broaden our understanding of the quality of life of nurses in pandemic contexts. Your perspective is valuable, and we will keep it in mind as we continue to develop our research.
Reviewer 2 Report
Comments and Suggestions for Authors
Authors developed an interesting research about quality of life of Spanish nurses during the sixth wave of the covid-19 pandemic and its association with other components such the family and labor conditions.
This manuscript is well written and has an introduction that provides enough background to understand their objectives, also, it is supported by relevant references.
The methods of this manuscript are also clear. Why has the SWING Questionnaire been highlighted in bold?
The results are well explained, and the tables clarify these results. Abbreviations in all tables should be explained.
I have some suggestions to modify this manuscript on the discussion section. In my opinion, the section that corresponds to lines 325-335 should be better explained. The relationship between work-life balance policies, the number of children and quality of life is not well understood.
In the conclusion, there is a need extended and clearer implications for clinical practice. What parameters do the organisations have to work with, to make nurses to feel improved the life quality in relation to elements studied? Conclusions should be more correlated with the aims of the study.
I hope this helps
Author Response
Authors developed an interesting research about quality of life of Spanish nurses during the sixth wave of the covid-19 pandemic and its association with other components such the family and labor conditions.
This manuscript is well written and has an introduction that provides enough background to understand their objectives, also, it is supported by relevant references.
The methods of this manuscript are also clear. Why has the SWING Questionnaire been highlighted in bold?
Response:
We sincerely appreciate your detailed feedback on our work. Regarding the highlighted text in bold in the SWING Questionnaire, we acknowledge that it was a typographical error which we have corrected in the revised version of the manuscript. We apologize for any confusion this may have caused and thank you for bringing it to our attention.
Principio del formulario
Final del formulario
The results are well explained, and the tables clarify these results. Abbreviations in all tables should be explained.
Response:
We want to thank you for pointing it out. We have also taken into account your suggestion regarding the abbreviations in the tables and have made the necessary modifications to include explanations for all the abbreviations used. We firmly believe that this improvement will significantly enhance the clarity and understanding of the data presented in our work.
I have some suggestions to modify this manuscript on the discussion section. In my opinion, the section that corresponds to lines 325-335 should be better explained. The relationship between work-life balance policies, the number of children and quality of life is not well understood.
Response:
Thank you for your valuable feedback on our manuscript. We appreciate your insightful suggestions regarding the discussion section, particularly regarding lines 325-335. We have carefully revisited this section and have made the necessary adjustments to provide a clearer explanation of the relationship between work-life balance policies, the number of children, and quality of life.
Your input has been instrumental in enhancing the clarity and coherence of our manuscript. We believe that the revisions we've made address the concerns you raised and contribute to a more comprehensive understanding of the topic.
In the conclusion, there is a need extended and clearer implications for clinical practice. What parameters do the organisations have to work with, to make nurses to feel improved the life quality in relation to elements studied? Conclusions should be more correlated with the aims of the study.
I hope this helps
Response:
Thank you for your helpful feedback regarding about the conclusion section of our manuscript. We acknowledge the need for extended and clearer implications for clinical practice, particularly in relation to the parameters organizations can utilize to enhance nurses' quality of life based on the elements studied.
In response to your suggestions, we have revised the conclusion to ensure stronger correlation with the aims of the study and provide more explicit guidance for clinical practice. We have emphasized specific parameters and strategies that organizations can implement to improve nurses' quality of life, aligning closely with the objectives outlined in the study.
Thank you for your valuable input and guidance throughout the review process.
Reviewer 3 Report
Comments and Suggestions for Authors
* Are all the cited references relevant to the research? Many of the references are more than 5-10 years old. A few are within the five-year range; however, I suggest finding more current references and then updating citations. 43/72 older than 5-10 years 60%
* Are the methods adequately described? The data collection period was only one month, Nov-Dec 2021. Short timeframe to attempt generalization. Snowball sampling may have contributed to a rushed study (participant response rates).
Title: Capitalize Spanish. Determine how will refer to COVID-19 or covid-19 and be consistent throughout manuscript.
Purpose: well-articulated.
Line 103 & 105: if you used both types of sampling, mention why
Line 166-167: how did you determine the internal consistency?
Line 190-195: would it be possible to label some of the tables with these tests? Or actually show the tables for the readers?
Line 198: Is the ‘25’ the version of SPSS? If so, please annotate. For platforms other than Windows, substitute the name of the operating system. For example: IBM SPSS Statistics for Macintosh, Version 28.0.
Line 232: what does family conciliation or reconciliation mean? Did you define this somewhere else?
Line 308: carers? Of whom, need to specify concisely for clarity.
Line 322: please explain where you are referring to the multivariate analysis, from your study or another?
Line 359: may have to use page number when indicate an author ‘stated’ or rephrase the sentence
Line 373: in the citation/sentence, spell out & = and
Line 381: delete participants
Line 395: delete positive work-family
Comments on the Quality of English LanguageCheck noun – verb tense.
Author Response
REVISOR 3
* Are all the cited references relevant to the research? Many of the references are more than 5-10 years old. A few are within the five-year range; however, I suggest finding more current references and then updating citations. 43/72 older than 5-10 years 60%
Response:
We appreciate your observation, and we have updated our bibliography with more current references that closely align with the scope and objectives of our research. In this way, we aim to ensure that our study is grounded in recent and relevant academic work available.
* Are the methods adequately described? The data collection period was only one month, Nov-Dec 2021. Short timeframe to attempt generalization. Snowball sampling may have contributed to a rushed study (participant response rates).
Response:
We sincerely appreciate your observation regarding the duration of the data collection. The decision to limit this period to one month was based on practical and contextual considerations. Firstly, one month provides an efficient timeframe to gather a significant amount of data without unnecessarily prolonging the study. Additionally, this shorter period helps minimize potential variations introduced by changes in the environment or participants' circumstances.
Secondly, a one-month data collection period is more feasible and manageable for both researchers and study participants. We consider it important to maintain adequate logistics and ensure active participation from those involved within a reasonable timeframe.
Lastly, in the context of a pandemic, conditions can change rapidly. One month can offer an opportunity to capture a snapshot of participants' experiences that is relevant and useful for decision-makers.
We acknowledge that a longer data collection period could provide a more comprehensive insight into the situation. However, we believe that one month is an appropriate and justifiable timeframe for the objectives and scope of this study. We appreciate your feedback and remain open to any additional suggestions that may enhance the quality of our research.
Title: Capitalize Spanish. Determine how will refer to COVID-19 or covid-19 and be consistent throughout manuscript.
Response:
We appreciate your observation, and we have made the necessary changes.
Purpose: well-articulated.
Line 103 & 105: if you used both types of sampling, mention why
Response: Actually, the study participants were recruited through snowball sampling rather than convenience sampling. We apologize for the confusion and acknowledge that it was a mistake to describe it that way. We have removed that reference from the manuscript's method section and will ensure that the description is accurate in the revised version. Thank you for pointing out this aspect.
Line 166-167: how did you determine the internal consistency?
Response:
We apologize for the misunderstanding. Internal consistency was determined using Cronbach's alpha coefficient, which has been included in the manuscript. Thank you for bringing this to our attention.
Line 190-195: would it be possible to label some of the tables with these tests? Or actually show the tables for the readers?
Response:
Thank you for suggesting this change. We have modified the table titles, and we hope they are clearer.
Line 198: Is the ‘25’ the version of SPSS? If so, please annotate. For platforms other than Windows, substitute the name of the operating system. For example: IBM SPSS Statistics for Macintosh, Version 28.0.
Response:
Thank you for your inquiry. We have made the necessary changes according to your instructions.
Principio del formulario
Final del formulario
Line 232: what does family conciliation or reconciliation mean? Did you define this somewhere else?
Response:
Thank you for bringing this to our attention. It was a typographical error, and we have corrected it in the manuscript.
Line 308: carers? Of whom, need to specify concisely for clarity.
Response:
Thank you for your feedback. It was a translation error, and we have corrected it by replacing "carers" with "the ones cared for" in the manuscript.
Line 322: please explain where you are referring to the multivariate analysis, from your study or another?
Response:
Thanks for your indication, this refers to our multivariate analysis. We have amended the text accordingly.
Line 359: may have to use page number when indicate an author ‘stated’ or rephrase the sentence
Response:
Thank you for your suggestion, we have added the year.Principio del formulario
Final del formulario
Line 373: in the citation/sentence, spell out & = and
Thank you for bringing this to our attention. It was a typographical error, and we have corrected it in the manuscript.
Line 381: delete participants
Thank you for bringing this to our attention. It was a typographical error, and we have corrected it in the manuscript.
Line 395: delete positive work-family
Thank you for bringing this to our attention. It was a typographical error, and we have corrected it in the manuscript.
We want to express our sincere gratitude for dedicating your time and effort to review our manuscript. Your comments and suggestions have been extremely valuable in improving the quality and clarity of our work. We deeply appreciate your dedication and attention to detail, which has significantly contributed to strengthening our study.
Reviewer 4 Report
Comments and Suggestions for Authors
Thank you authors for manuscript. I believe that important revisions need to be applied
INTRODUCTION
-Insert the 2011 WHO health definition proposal
-The stigmatization of such healthcare workers in the immediate pandemic period (write about this)
-In our opinion, studies that have explored how the pandemic has influenced QoL are present but certainly not as specific.
METHOD
-I ask the authors to specify the inclusion criteria in the study....is it enough to be a nurse?
-The Work-Family Interaction Questionnaire (SWING) is psychometrically validated for this population
ETHICAL
-Protocol approval date missing
DISCUSSIONS
EMPHASIZE THE UNIQUENESS OF THE STUDY
Comments on the Quality of English Languageminor editing
Author Response
Thank you authors for manuscript. I believe that important revisions need to be applied
INTRODUCTION
-Insert the 2011 WHO health definition proposal
Response:
We appreciate your suggestion to include the proposed definition of health from the WHO in 2011. However, after a thorough review of the literature, we did not find any official modification to the WHO definition of health in 2011. Although there were debates in that year regarding the need to update the definition to reflect the rise of chronic diseases, the official definition has not been modified since 1948.
Therefore, we have reformulated the line in question to reflect this information: "Health status and health-related quality of life (HRQoL) have been used interchangeably since the World Health Organization (WHO) defined health in 1948. Despite debates in 2011 about the need to update this definition to reflect the increase in chronic diseases, the official definition has not been modified."
We hope this modification is satisfactory, and we are open to further suggestions to improve the manuscript.
-The stigmatization of such healthcare workers in the immediate pandemic period (write about this)
Response:
We appreciate your feedback, and we have added a paragraph on this topic in the introduction. In lines 78-86, we discussed the stigmatization of healthcare workers during the immediate pandemic period.
-In our opinion, studies that have explored how the pandemic has influenced QoL are present but certainly not as specific.
Response:
We agree with that statement. In fact, our team has conducted a review on health-related quality of life (HRQoL), which is still pending publication. In this review, we observed that there are few articles specifically addressing HRQoL. Most of the studies found focus on aspects related to mental health, burnout, and professional quality of life, rather than health-related quality of life.
METHOD
-I ask the authors to specify the inclusion criteria in the study....is it enough to be a nurse?
Response:
We appreciate your question regarding our inclusion criteria. Deliberately, we kept our criteria broad, requiring only that participants were nurses. Our aim was to embrace a diverse range of nursing professionals to capture a wide array of perspectives and experiences related to our research objectives. Therefore, being a nurse was the primary criterion for inclusion in our study. Moreover, all participants were registered with Spanish professional nursing associations and actively engaged in their profession during the data collection period.
-The Work-Family Interaction Questionnaire (SWING) is psychometrically validated for this population
Response:
Indeed, the Work-Family Interaction Questionnaire (SWING) is psychometrically validated for the Spanish population and demonstrates good internal consistency, with Cronbach's alpha values ranging between 0.77 and 0.89.
ETHICAL
-Protocol approval date missing
Response:
Thank you for your suggestion. We have included the protocol approval date in the manuscript.
DISCUSSIONS
EMPHASIZE THE UNIQUENESS OF THE STUDY
Response:
Following your guidance, we have added a paragraph in the discussion section and modified the conclusion to emphasize the originality and relevance of our research.
We appreciate all your comments and remain available for any further suggestions or feedback you may have to enhance our work.
Round 2
Reviewer 3 Report
Comments and Suggestions for Authors
COVID-19 VS. covid-19, please correct for consistency throughout the manuscript.
Line 74: Anthropomorphism
Line 115 & 122: "quotation" vs. single 'quote marks': determine and revise for consistency
Line 172: You may need to reword how you validated the questionnaire you created for administration to your study participants.
Table 1: close contact INSULATION? Is this a typo that maybe is Isolation?
Line 240 & Table 2: conciliation? What does this mean? Please clarify and make amendments to the rest of the manuscript for consistency.
Great follow-up with the strengths of this study. Much better conclusion. Updating the references is evident in the body of this manuscript, great job.
Comments on the Quality of English Language
Typos evident, suggested corrections in author comments.
Author Response
COVID-19 VS. covid-19, please correct for consistency throughout the manuscript.
Response:
We would like to extend our apologies for not correctly implementing the requested change in the previous review regarding the consistent use of COVID-19 throughout the manuscript. We sincerely appreciate you bringing this error to our attention.
Immediate action has been taken to rectify the situation, and necessary changes have been made to ensure all references to COVID-19 are uniform throughout the text. We hope that we have addressed the request appropriately this time.
Thank you for your patience and continued attention to detail. Your review and guidance are instrumental in improving the quality of the manuscript. We are grateful for your collaboration and commitment to this process.
Line 74: Anthropomorphism
Response:
Thank you for your observation regarding Line 74 and the mention of 'Anthropomorphism,' attributing human characteristics to science. We appreciate your keen attention to detail.
We acknowledge that this was an error in translation, and we have promptly amended the sentence to reflect the intended meaning: 'the lack of previous scientific knowledge in this area caused significant difficulties in most countries worldwide.'
Line 115 & 122: "quotation" vs. single 'quote marks': determine and revise for consistency
Response:
Thank you for bringing to our attention the inconsistency in the use of quotation marks between "quotation" and single 'quote marks' in lines 115 and 122. We have proceeded with the necessary changes and revised the manuscript to ensure consistency throughout, using double quotation
Line 172: You may need to reword how you validated the questionnaire you created for administration to your study participants.
Response:
Thank you for your comment. We have modified the line to include information regarding the population for which the questionnaire was validated and the authors who conducted the validation process
Table 1: close contact INSULATION? Is this a typo that maybe is Isolation?
Response:
We appreciate you bringing this to our attention so we can correct it accordingly.
Line 240 & Table 2: conciliation? What does this mean? Please clarify and make amendments to the rest of the manuscript for consistency.
Response:
We have amended "conciliation" to "work-life balance" throughout the manuscript to enhance clarity and consistency. Thank you for highlighting this issue
Great follow-up with the strengths of this study. Much better conclusion. Updating the references is evident in the body of this manuscript, great job.
Response:
Thank you for your encouraging feedback. We truly appreciate all the comments and suggestions provided in both this and the previous reviews. Your valuable insights have significantly contributed to the improvement of the manuscript. Finally, we would like to inform you that the manuscript has been sent, once again, to our translator to enhance the level of English language proficiency. Your observations regarding the need for improvement in this aspect are duly noted, and we are committed to ensuring the highest quality of language in our work. She is an official translator from English into Spanish and from Spanish into English affiliated with the International Association of Translators, Proofreaders, and Interpreters, with the membership number shown on the attached certificate.
Reviewer 4 Report
Comments and Suggestions for Authors
thank to authors for responding to all the comments
Comments on the Quality of English Languageminor editing
Author Response
thank to authors for responding to all the comments
Response:
We would like to extend our most sincere gratitude for all the comments that have contributed to improving the quality of the manuscript. Your feedback has been invaluable in this process. We would also like to add that before submitting the article, we sent it to our translator to address concerns about the quality of the English language. She is an official translator from English into Spanish and from Spanish into English affiliated with the International Association of Translators, Proofreaders, and Interpreters, with the membership number shown on the attached certificate. We hope this is correct.